# Comparison of Glucose Lowering Efficacy of Human GLP-1 Agonist in Taiwan Type 2 Diabetes Patients after Switching from DPP-4 Inhibitor Use or Non-Use

**DOI:** 10.3390/jpm12111915

**Published:** 2022-11-16

**Authors:** Chia-Jen Tsai, Cheng-Feng Tsao

**Affiliations:** Division of Endocrinology and Metabolism, Department of Internal Medicine, Kaohsiung Chang Gung Memorial Hospital and Chang Gung University, College of Medicine, Kaohsiung 833, Taiwan

**Keywords:** GLP-1 agonist, DPP-4 inhibitor, type 2 diabetes mellitus, Taiwan

## Abstract

To determine the efficacy of glucose control in type 2 diabetes patients who switch from dipeptidyl peptidase-4 (DPP-4) inhibitors use or non-use to GLP-1 receptor agonists (GLP-1 RAs). We conducted a cohort study using data from the Chang Gung Research Database. Patients aged ≥18 years using newly initiated GLP-1 RAs between 1 January 2009, and 31 December 2016, were included. Cox proportional hazards models were used to adjust for treatment selection bias. The primary outcome was changes in the glycated hemoglobin (HbA1c) level. The HbA1c level fell substantially after initiating GLP-1 RAs in DPP-4 inhibitor users and nonusers. A mean HbA1c reduction of −0.42% was found in patients who received DPP-4 inhibitors. Those who were DPP-4 inhibitor nonusers had a reduction in HbA1c of −0.99%. The degree of reduction in HbA1c was significantly greater in patients who were DPP-4 inhibitor nonusers (*p* value < 0.01), compared to the DPP-4 inhibitor users. In routine care, DPP-4 inhibitor nonusers had better efficacy in glucose control than DPP-4 inhibitor users after switching to a GLP-1 agonist.

## 1. Introduction

Incretin therapy is now recommended as a second-line treatment for type 2 diabetes, according to the newest guidelines [1,2]. GLP-1 receptor agonists (GLP-1 RAs) act via activation of the GLP-1 receptor [3], and dipeptidyl peptidase-4 (DPP-4) inhibitors act by increasing levels of active incretin hormone. Both GLP-1 RAs and DPP-4 inhibitors have well-established antidiabetic activity compared to a placebo [4,5,6,7,8]. However, some differences exist between GLP-1 RAs and DPP-4 inhibitors. For example, GLP-1 RAs have demonstrated the ability to induce significant weight loss in randomized control trials and real-world evidence [7,8,9,10,11], especially with the new molecules of the GLP-1 RA [12].

For those with impaired glucose tolerance, a decrease in the GLP-1 concentration was found after glucose challenge [13]. Several double-blind, randomized controlled, or active comparator trials have found that GLP-1 RAs yield a greater reduction in HbA1c than DPP-4 inhibitors. DeFronzo and colleagues found that subjects who were administered exenatide had a significant reduction in 2-h postprandial glucose [14]. A multicenter, open-label, parallel group study showed liraglutide 1.2 doses of mg and 1.8 mg displayed superiority in reducing HbA1c compared to sitagliptin [15]. Furthermore, GLP-1 RAs reduced the risk of cardiovascular and renal adverse events in several large cardiovascular outcome trials [7,8,9]. Because of their better efficacy and cardiovascular outcome, GLP-1 RAs have recently been recommended for patients with metformin treatment failure and evidence of cardiovascular disease. However, there is little research on the efficacy of glucose control after switching to GLP-1 RAs in those who received DPP4 inhibitors but failed. Our study aimed to examine the effect of GLP-1 RAs in cases of treatment failure.

## 2. Methods

### 2.1. Data Source

The Chang Gung Research Database (CGRD) is the largest multi-institutional electronic medical record database at Chang Gung Memorial Hospital (CGMH) in Taiwan and includes two medical centers, two regional hospitals, and three district hospitals in Taiwan. The CGRD contains patient-level data derived from electronic medical charts of patients established for administrative and health care purposes at CGMH. All electronic medical records from CGMH were integrated and standardized in the database, which was established for research purposes. The database contains all information on inpatient admissions, outpatient services, outpatient prescription drug dispensing, laboratory data, and disease category data. Disease diagnoses registered in the CGRD are classified according to the International Classification of Disease, Ninth Revision, Clinical Modification (ICD-9-CM).

### 2.2. Study Population

We identified adult patients with diabetes, according to ICD-9-CM codes 580.x, and 584.x, in the CGRD. Figure 1 depicts the algorithm used for patient selection. The study participants were selected from CGMH data for 1 January 2009, to 31 December 2016. The definition of diabetes in this cohort was based on the following criteria: having visited at least three outpatient clinics where they were diagnosed with diabetes (ICD-9-CM code 250.x). Patients were excluded from the cohort if they (1) were <18 years old on the index data, (2) had received GLP-1 RAs less than 3 times (so that only patients are included: (1) who can tolerate the drug, (2) who are stable users are included), or (3) lacked the available laboratory data. The GLP-1 RAs included exenatide, liraglutide and dulaglutide. The DPP-4 inhibitors were sitagliptin, vildagliptin, saxagliptin, linagliptin and alogliptin.Statistical HbA1c change should be collected at least 3 months after switching drugs.

### 2.3. Statistical Analyses

The characteristics of DPP-4 inhibitor nonusers and users were compared using the Student’s *t* test for age and the chi-square test for other variables.

Statistical analyses were performed using SAS version 25 (SAS Institute, Cary, NC, USA). *p* ≤ 0.05 is considered significant.

## 3. Results

### 3.1. Characteristics of the Study Population

In all, 963 patients with a diagnosis of diabetes and who received GLP-1 RAs between 1 January 2009, and 31 December 2016, were identified and recruited for this study (Figure 1). Included in this study group were 754 patients who received DPP-4 inhibitors before starting GLP-1 RAs and 209 patients who were DPP-4 inhibitor nonusers. The DPP-4 inhibitor users stop that treatment when starting on GLP-1 RAs according to the restriction of Taiwan’s Bureau of National Health Insurance.

The demographic and clinical characteristics of these cohorts were estimated and are summarized in Table 1. The mean HbA1c of all patients was 9.34%. The percent of patients with HbA1c less than 7 was 2.91%. Those patients who received three different types of antidiabetic drugs accounted for 22.22% of all patients, and those who received more than four kinds of drugs accounted for 62.62%.

DPP-4 inhibitor users, compared to DPP-4 inhibitor nonusers, had less hypercholesterolemia and LDL but higher serum creatinine. The DPP-4 inhibitor user group was more likely to have received more than four different types of oral anti-hyperglycemia agents (74.54%). However, most of the DPP-4 inhibitor nonusers (54.06%) had received two or three kinds of oral anti-hyperglycemic agents. The proportion of HbA1c greater than 9 was 57.43% in the DPP-4 inhibitor user group and 60.29% in the DPP-4 inhibitor nonuser group, respectively.

### 3.2. Glucose Control Efficacy of Switching to GLP-1 RAs

Improvements in HbA1c were found in all patients after switching to GLP-1 RAs (Table 2): the mean HbA1c decreased 0.54% regardless of the previous oral anti-hyperglycemic agents used. Patients who received DPP-4 inhibitors exhibited a significant reduction in HbA1c (−0.42, 95%CI −3.05% to 1.07%). DPP-4 inhibitor nonusers who switched also sustained improvement in HbA1c (−0.99, 95%CI −3.05% to 1.07%). The degree of reduction in HbA1c was significant among DPP-4 inhibitor nonusers (*p* value < 0.01), compared to DPP-4 inhibitor users. The improvement in HbA1c was more significant for patients with HbA1c > 9.0% in both groups. Among those with HbA1c between 7 and 9, the DPP-4 inhibitor nonusers had significantly better blood glucose control, compared to DPP-4 inhibitor users. The amount of oral anti-hyperglycemic agent used before switching to GLP-1 RAs had no effect in both groups.

Among DPP-4 inhibitor users, those who initially had received more than three different oral anti-hyperglycemia agents had a significant reduction in mean HbA1c after switching to GLP-1 RAs. For those who received 3 and more than four oral anti-hyperglycemic agents before switching, the mean HbA1c was significantly reduced by −0.91% and −0.27%, respectively. DPP-4 inhibitor users who had a baseline HbA1c greater than or equal to 9 could have a significant reduction in HbA1c after switching to GLP-1 RAs. However, the mean HbA1c increased for those with a baseline HbA1c less than 9 after switching to GLP-1 RAs.

For DPP-4 inhibitor nonusers, the mean HbA1c reduction was significant in those who initially received fewer than four different oral anti-hyperglycemic agents. There was a reduction in HbA1c (−1.81%) among those with a baseline HbA1c greater than or equal to 9.

## 4. Discussion

In the present study, we estimated and compared the effects of oral anti-hyperglycemic agents on HbA1c after switching to GLP-1 RAs in Taiwan primary care. In cohorts of people with similar baseline characteristics and levels, we found better reductions in HbA1c in DPP-4 inhibitor nonusers than in DPP-4 inhibitor users.

There is a change in circulating incretin levels in prediabetic patients. In most patients with prediabetes, a decrease in the GLP-1 concentration was found after glucose challenge, and this was not related to a deficiency in first-stage basal insulin secretion [13]. Therefore, a GLP-1 decrease was found mainly in people with impaired glucose tolerance, who are already known to have a state of insulin resistance [16,17]. In addition, it was found that early glucagon inhibition is impaired in those with impaired glucose tolerance [18].

The effect of GLP-1 RAs on patients with diabetes has been demonstrated in multiple large trials. The mean HbA1c fell by 0.9% in subjects with exenatide use [19], and by 1.48% in subjects with liraglutide use [19]. Different classes of GLP-1 agonists had similar efficacy with regard to glucose control. In our study, we found that after switching to GLP-1 RA, there was a mean HbA1c reduction of −0.54%, regardless of the previous oral anti-hyperglycemia agents used. In the AMIGO trials evaluating exenatide in type 2 diabetes inadequately controlled, a reduced mean HbA1c of 0.5% with 5 mg BID exenatide [20,21,22] was found. In the LEADER trial [7], a decreased mean HbA1c of −0.40% was found with liraglutide use compared with the placebo group. The REWIND study showed a HbA1c reduction of 0.61% in the dulaglutide group, compared with the placebo group [9]. In the SUSTAIN 6 study [8], the mean HbA1c level in the semaglutide group was 0.7% lower in the group receiving 0.5 mg and 1% lower in the group receiving 1.0 mg semaglutide, compared with the placebo group. The efficacy of glycemic control in our study was similar to that of previous large trials.

The SUSTAIN 2 trial was a 56-week randomized, double-blind, multinational, multicenter trial, and eligible patients were diagnosed with type 2 diabetes with insufficient glycemic control. The baseline mean HbA1c was 8.1%. The SUSTAIN 2 trial found a mean HbA1c reduction of 1.3% in the semaglutide 0.5 mg group, 1.6% in the semaglutide 1.0 mg group and 0.5% in the sitagliptin group, respectively [23]. In a 78-week open-label trial, those who were treated with sitagliptin had additional changes of −0.2~−0.5% in HbA1c after switching to liraglutide [24]. However, participants who received sitagliptin added to metformin had improvements in glycemic control for 52 weeks. The DURATION-2, a subsequent study, demonstrated significant incremental improvement in HbA1c (−0.3%) in those who switched from sitagliptin to exenatide once weekly [25]. The mean baseline HbA1c was 8.5% in the DURATION-2 study. In our study, the baseline mean HbA1c was 9.34%, higher than that of eligible patients in most trials. Most of the subjects in our study had HbA1c levels greater than 9 and had treatment failure. Our study found a mean HbA1c reduction of 0.54% after switching to GLP-1 RAs. Besides, those who received a DPP-4 inhibitor first had a poor response regarding glucose control after switching to GLP-1 RAs in our study.

Our study was the first to report that the benefit of glucose control was less predominant in DPP-4 inhibitor users than in nonusers before switching to GLP-1 RAs. Our study population included individuals with diabetes with higher HbA1c than in other studies, which meant that most patients still had poor glycemic control even after receiving DPP-4 inhibitors. Taiwan’s Bureau of National Health Insurance has stipulated that GLP-1 RAs should be added after metformin and/or sulfonylurea treatment failure. However, Taiwanese are hesitant about using the injection form of GLP-1 RAs. This leads to patients having higher HbA1c levels when receiving GLP-1 RAs.

Current evidence suggests that higher baseline HbA1c is associated with greater efficacy of both DPP-4 inhibitors and GLP-1RA therapies in lowering HbA1c [26]. This study only showed significantly decreased HbA1c in the patients with baseline HbA1c > 9 when receiving GLP-1 RA.

DPP-4 inhibitors increase active GLP-1 concentrations by 2 or 3 times the concentration at baseline. However, the stimulation of GLP-1 receptor activity with GLP-1RA is several times higher than with DPP-4 inhibitors [27,28].

We speculate that, compared with the DPP-4 inhibitor-exposed patient, DPP-4 inhibitors naïve patient could have more active GLP-1 concentrations increasing, after adding GLP-1RA, which may be the reason for the difference in HbA1c.

Since type 2 diabetes is considered a progressive disease with slow loss of beta cell function, the waning effect of each medication was documented, and the projected time to A1c neutrality was documented over the ADOPT trial [29]. The pretreated A1C levels have a modest effect on the fall of A1C levels in response to treatment [30]. Patients pre-treated with a DPP-4 inhibitor were treated with more kinds of antidiabetic drugs than the DPP-4 inhibitor naïve patient in our study. The patient with less glucose reduction after receiving injection of GLP1-RA should have longstanding diabetes.

For those who received DPP-4 inhibitors and still had poor glycemic control, incretin therapy appears to be less effective. Early use of GLP-1 may be a better choice for those who have poor glycemic control and have never received incretin therapy.

Our study has some limitations that should be acknowledged. First, with the impossibility of treatment randomization in this retrospective and observational study, the influence of potential confounding factors not evaluated herein may have biased the results. For example, the renal function or percentage of insulin may be related to the duration of the disease, which is practical information for clinicians to start GLP1, but some baseline characteristics of the study patients (e.g., fasting serum glucose and duration of diabetes) were not available in the CGRD. Second, our database was derived from a large institutional electronic medical records database in Taiwan. The selection biases, which may differ from those of the national database, should be evaluated. Third, some unstructured information that is recorded in the original electronic medical record but is currently not available in the CGRD, for example, physician’s notes, such as subjective, objective, assessment, plan and some variables (including follow-up body weight).

However, it was found that the CGRD could serve as the basis for accurate estimates [31].

Our findings suggest that patients who did not receive DPP-4 inhibitors had better glucose control after switching to GLP-1 RAs. These findings may shed further light on the management of diabetes care, which may aid in making treatment decisions in routine clinical practice.

## Figures and Tables

**Figure 1 jpm-12-01915-f001:**
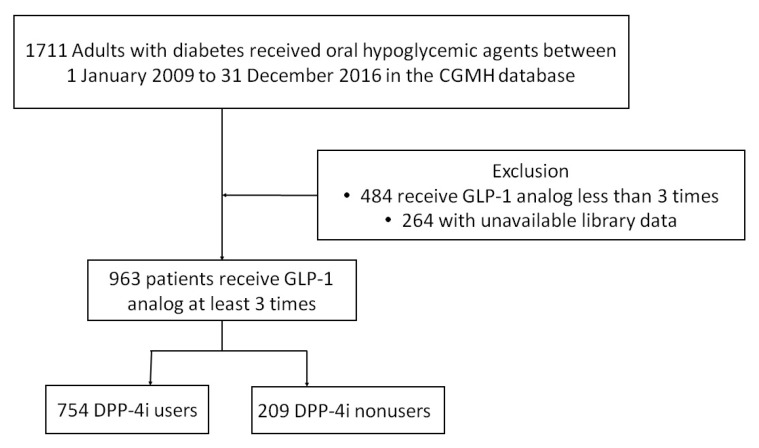
Flow diagram for selecting study patients. The study participants selected from CGRD records from 1 January 2009 to 31 December 2016 were screened using inclusion and exclusion criteria. A total of 963 patients were identified for the final analysis.

**Table 1 jpm-12-01915-t001:** Baseline characteristics of the study population before switching to GLP-1 RAs. Data are mean (SD), median (range) or *n* (%).

	DPP4-i Users	DPP4-i Nonusers	*p* Value
	(*n* = 754)	(*n* = 209)	
Age (years)	51.7 (11.5)	50.9 (12.1)	0.369
Sex: male (%)	336 (44.5%)	95 (45.5%)	0.818
Height (cm)	163.1 (8.7)	162.4 (9.6)	0.302
Weight (kg)	83.4 (17.8)	84.6 (20.9)	0.452
Body mass index (kg/m^2^)	31.3 (5.6)	31.9 (6.3)	0.300
HbA1c (%)	9.3 (1.5)	9.3 (1.4)	0.873
Total cholesterol (mg/dL)	174.6 (33.9)	183.6 (38.0)	0.001
Triglycerides (mg/dL)	194.8 (148.9)	194.5 (144.52)	0.981
HDL-cholesterol (mg/dL)	43.7 (11.1)	45.2 (10.1)	0.087
LDL-cholesterol (mg/dL)	97.6 (26.2)	106.6 (29.6)	<0.001
GPT (U/L)	40.2 (29.3)	42.5 (35.0)	0.414
Creatinine (mg/dL)	0.93 (0.78)	0.83 (0.30)	0.010
Number of oral anti-hyperglycemia agents			
Metformin	740 (98.1%)	170 (81.3%)	<0.001
Sulfonylurea	700 (92.8%)	134 (64.1%)	<0.001
Glinides	80 (10.6%)	10 (4.8%)	0.01
Thiazolidinedione	367 (48.7%)	56 (26.8%)	<0.001
α-glucosidase inhibitors	338 (44.8)	44 (21.1%)	<0.001
SGLT2 inhibitors	2 (0.3%)	0 (0.0%)	1.000
Insulin regimen			
Basal insulin	284 (37.7%)	60 (28.7%)	0.017
Rapid-acting insulin	46 (6.1%)	18 (8.61%)	0.197
Premix insulin	203 (26.9%)	55 (26.3%)	0.861

**Table 2 jpm-12-01915-t002:** Changes in HbA1c by treatment group. Data are mean ± SD.

	DPP-4 i Users				DPP-4 I Nonusers				DPP4-i Users	DPP4-i Nonusers	
		Before GLP-1 RAs	After GLP-1 RAs			Before GLP-1 RAs	After GLP-1 RAs		Change from baseline	Change from baseline	
Glycemic outcome	N	HbA1c	HbA1c	*p*-value	N	HbA1c	HbA1c	*p*-value	HbA1c	HbA1c	*p*-value
	754	9.34 (±1.48)	8.92 (±1.42)	<0.001	209	9.32 (±1.40)	8.34 (±1.55)	<0.001	−0.42 (±2.04)	−0.99 (±2.06)	<0.001
Number of diabetes medications											
0	5 [0.66%]	8.36 (±1.26)	7.80 (±1.27)	0.312	25 [11.96%]	9.11 (±1.31)	8.13 (±1.62)	0.007	−0.56 (±1.08)	−0.98 (±1.67)	0.594
1	5 [0.66%]	9.56 (±0.91)	9.02 (±1.09)	0.363	30 [14.35%]	9.25 (±1.35)	7.55 (±1.13)	<0.001	−0.54 (±1.18)	−1.69 (±1.80)	0.178
2	24 [3.18%]	9.18 (±1.32)	8.43 (±1.31)	0.064	57 [27.27%]	9.45 (±1.40)	8.25 (±1.38)	<0.001	−0.75 (±1.88)	−1.20 (±2.08)	0.360
3	158 [20.95%]	9.46 (±1.58)	8.55 (±1.36)	<0.001	56 [26.79%]	9.18 (±1.44)	8.39 (±1.62)	0.013	−0.91 (±2.22)	−0.79 (±2.31)	0.743
≥4	562 [74.53%]	9.32 (±1.46)	9.05 (±1.42)	0.001	41 [19.61%]	9.52 (±1.46)	9.08 (±1.64)	0.157	−0.27 (±1.99)	−0.44 (±1.97)	0.586
Group by level of HbA1c											
<7	23 [3.05%]	6.49 (±0.38)	8.93 (±1.41)	<0.001	5 [2.39%]	6.44 (±0.50)	7.36 (±0.81)	0.081	2.44 (±1.40)	0.92 (±0.89)	0.029
7~9	298 [39.52%]	8.13 (±0.52)	9.00 (±1.44)	<0.001	78 [37.32%]	8.11 (±0.54)	8.33 (±1.49)	0.200	0.87 (±1.53)	0.22 (±1.51)	0.001
≧9	433 [57.42%]	10.33 (±1.08)	8.87 (±1.41)	<0.001	126 [60.28%]	10.19 (±1.04)	8.38 (±1.60)	<0.001	−1.46 (±1.70)	−1.81 (±1.98)	0.052

## Data Availability

The data that support the findings of this study are available from the Chang Gung Research Database but restrictions apply to the availability of these data, which were used under license for the current study, and so are not publicly available. Data are however available from the authors upon reasonable request and with permission from the Chang Gung Research Database.

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
