# Peer review of "Comparison of Glucose Lowering Efficacy of Human GLP-1 Agonist in Taiwan Type 2 Diabetes Patients after Switching from DPP-4 Inhibitor Use or Non-Use"

_jpm, 2022, doi:10.3390/jpm12111915_

Round 1
Reviewer 1 Report
this paper is well written and data are interesting for the novelty of RWE in diabetic treatement. unfortuently data of outcomes are not exaustive.
in the paper Clinical efficacy and predictors of response to dulaglutide in type-2 diabetes by . Berra ant al on pharmacol res 2020 (Sep;159:104996) similar data were reported on 104 subjects with IDPPIV treatment with a lowering of HbA1c of 0.8% but with lower Hba1c basal data. moreover It's important to notice in this paper the lowering of BMI not described in your work. It's possibile to have there data? another important notice could be the subgroup analisys of the different GLP1 used
I suggest to add also in the references in the intro for the weight loss the review from Anti-diabetic drugs and weight loss in patients with type 2 diabetes. on Pharmacol Res. 2021 Sep;171:105782. doi:
and for another RWE study the effect of GLP1 liraglutide on CVR by Mirani et al. Liraglutide and cardiovascular outcomes in a real world type 2 diabetes cohort. on Pharmacol Res. 2018 Sep 10
Author Response
Please also see the attachment
Comments and Suggestions for Authors
Reviewer #1:
this paper is well written and data are interesting for the novelty of RWE in diabetic treatement. unfortuently data of outcomes are not exaustive.
in the paper Clinical efficacy and predictors of response to dulaglutide in type-2 diabetes by . Berra ant al on pharmacol res 2020 (Sep;159:104996) similar data were reported on 104 subjects with IDPPIV treatment with a lowering of HbA1c of 0.8% but with lower Hba1c basal data. moreover It's important to notice in this paper the lowering of BMI not described in your work. It's possibile to have there data? another important notice could be the subgroup analisys of the different GLP1 used
I suggest to add also in the references in the intro for the weight loss the review from Lazzaroni E et al Anti-diabetic drugs and weight loss in patients with type 2 diabetes. on Pharmacol Res. 2021 Sep;171:105782. doi:
and for another RWE study the effect of GLP1 liraglutide on CVR by Mirani et al. Liraglutide and cardiovascular outcomes in a real world type 2 diabetes cohort. on Pharmacol Res. 2018 Sep 10
Reply: Thanks for your suggestions. We agreed that efficacy of weight change on different GLP-1 RA is important, especially in particular the new molecules of the GLP-1 RA. At this point we do not have the data due to the limitation of data source, which from the Chang Gung Research Database (CGRD). It is an electronic medical record (EMR) database. There is some unstructured information that recorded in the original EMR but currently not available in the CGRD; for example, physician's notes, such as subjective, objective, assessment, plan and some variables (included follow-up body weight).
We addressed this issue at the limitation section.
We had also reviewed the references and added in the introduction for the weight loss as your suggestion to enrich the manuscript.

Reviewer 2 Report
The manuscript entitled ”Comparison of glucose lowering efficacy of human GLP-1 agonist in Taiwan type 2 diabetes patients after switching from DPP-4 inhibitor use or non-use” is showing that both DPP-4 users and DPP-4 nonusers lower the HbA1c levels after switching to GLP-1 agonist, but that the effect was bigger in the DPP-4 nonusers. This is as far as I know the first paper comparing the effect of GLP-1 agonists between former DPP-4 users and non-users. The manuscript has a clear structure and is overall well written, but I have a few comments and points that could be clarified.
Comments:
1. Page 2, Line 62: The sentence “Patients were excluded from the cohort if they (1) were <18 years on the index data, (2) had received GLP-1 RAs less than 3 times(so that only patients are included: (1)who can tolerate the drug, (2)they are stable users), or (3) lacked the available laboratory data.” is a little confusing since you are using two sets of numbers. I would suggest to take away the numbers 1 and 2 within the paranthesis to not confuse the reader and change to: “Patients were excluded from the cohort if they (1) were <18 years on the index data, (2) had received GLP-1 RAs less than 3 times (so that only patients are included: (1)who can tolerate the drug, (2)theyand are stable users are included), or (3) lacked the available laboratory data.”
2. Page 2, Line 80: Here it says “Included in this study group were 754 patients who received DPP-4 inhibitors before starting GLP-1 RAs and 209 patients who were DPP-4 inhibitor nonusers.” However, it’s not clearly stated if the DPP-4 inhibitor users did stop that treatment when starting on GLP-1 RA therapy. I also find it likely that those patients having other oral anti-diabetic treatments at study start keep their old medicines in combination with the GLP-1 RA treatment, but it’s not mentioned anywhere in the text. Please clarify.
3. Page 3, Line 93: The sentence “The proportion of HbA1c greater than 9 was - highest, with 57.43% in the DPP-4 inhibitor user group and 60.29% in the DPP-4 inhibitor nonuser group, respectively.” is a bit unclear for me. Was the proportion of patients with HbA1c greater than 9 highest in the DPP-4 inhibitor nonuser group (60.29% compared to 57.43% in the DPP-4 inhibitor user group)? Now it sounds like the opposite. Please clarify.
4. Page 4, Line 97: For me it’s unclear what the “after switching to GLP-1 RAs“ means in the sentence “Improvements in HbA1c were found in all patients after switching to GLP-1 Ras (Table 2)”. It says in the methods that those who had received GLP-1 RAs less than 3 times were excluded from the study, but it doesn’t say at which time point the after the switch the second data points were collected for the patients included in the study. Please clarify in the this in the methods section.
5. Page 4, Line 118: Could you comment upon the sentence “There was a significant reduction in HbA1c (-1.81%) among those with a baseline HbA1c greater than or equal to 9.” According to Table 2, the p-value for this statistical test was 0.052, and I can’t see any statement of which p-value is considered significant. Good to put in the methods section something about p ≤ 0.05 is considered significant.
6. Page 5, Line 169: In this paragraph you discuss that the efficacy of DPP-4 inhibitors and GLP-1 receptor agonists is greater if the baseline HbA1c is higher. Maybe it’s appropriate to mention here that this study only showed significantly decreased HbA1c in the patients with baseline HbA1c >9, when receiving GLP-1 RA.
Author Response
Please also see the attachment
Comments and Suggestions for Authors
Reviewer #2
The manuscript entitled ”Comparison of glucose lowering efficacy of human GLP-1 agonist in Taiwan type 2 diabetes patients after switching from DPP-4 inhibitor use or non-use” is showing that both DPP-4 users and DPP-4 nonusers lower the HbA1c levels after switching to GLP-1 agonist, but that the effect was bigger in the DPP-4 nonusers. This is as far as I know the first paper comparing the effect of GLP-1 agonists between former DPP-4 users and non-users. The manuscript has a clear structure and is overall well written, but I have a few comments and points that could be clarified.
Comments:
- Page 2, Line 62:The sentence “Patients were excluded from the cohort if they (1) were <18 years on the index data, (2) had received GLP-1 RAs less than 3 times(so that only patients are included: (1)who can tolerate the drug, (2)they are stable users), or (3) lacked the available laboratory data.” is a little confusing since you are using two sets of numbers. I would suggest to take away the numbers 1 and 2 within the paranthesis to not confuse the reader and change to: “Patients were excluded from the cohort if they (1) were <18 years on the index data, (2) had received GLP-1 RAs less than 3 times (so that only patients are included: (1)who can tolerate the drug, (2)theyand are stable users are included), or (3) lacked the available laboratory data.”
Reply: Thanks for your suggestions. This sentence was modified according to the comment.
- Page 2, Line 80:Here it says “Included in this study group were 754 patients who received DPP-4 inhibitors before starting GLP-1 RAs and 209 patients who were DPP-4 inhibitor nonusers.” However, it’s not clearly stated if the DPP-4 inhibitor users did stop that treatment when starting on GLP-1 RA therapy. I also find it likely that those patients having other oral anti-diabetic treatments at study start keep their old medicines in combination with the GLP-1 RA treatment, but it’s not mentioned anywhere in the text. Please clarify.
Reply: Thanks for your comments. Taiwan’s Bureau of National Health Insurance has stipulated that GLP-1 RAs should be added after metformin and/or sulfonylurea treatment failure and shouldn't combine with DPP-4 inhibitor, so the DPP-4 inhibitor users stop that treatment when starting on GLP-1 RA. Because it a retrospective and observational study, we can only analyze from prescriptions, and cannot know for sure whether the patient is using the original drug.
The revised section is added in this paragraph.
- Page 3, Line 93:The sentence “The proportion of HbA1c greater than 9 was - highest, with 57.43% in the DPP-4 inhibitor user group and 60.29% in the DPP-4 inhibitor nonuser group, respectively.” is a bit unclear for me. Was the proportion of patients with HbA1c greater than 9 highest in the DPP-4 inhibitor nonuser group (60.29% compared to 57.43% in the DPP-4 inhibitor user group)? Now it sounds like the opposite. Please clarify.
Reply: Thanks for your comments. The sentence was modified to “The proportion of HbA1c greater than 9 was 57.43% in the DPP-4 inhibitor user group and 60.29% in the DPP-4 inhibitor nonuser group, respectively.”
- Page 4, Line 97: For me it’s unclear what the “after switching to GLP-1 RAs“ means in the sentence “Improvements in HbA1c were found in all patients after switching to GLP-1 Ras (Table 2)”. It says in the methods that those who had received GLP-1 RAs less than 3 times were excluded from the study, but it doesn’t say at which time point the after the switch the second data points were collected for the patients included in the study. Please clarify in the this in the methods section.
Reply: Thanks for your comments. Statistical HbA1c change should be collected at least 3 months after switching drug. The revised section is now available in the methods section.
- Page 4, Line 118:Could you comment upon the sentence “There was a significant reduction in HbA1c (-1.81%) among those with a baseline HbA1c greater than or equal to 9.” According to Table 2, the p-value for this statistical test was 0.052, and I can’t see any statement of which p-value is considered significant. Good to put in the methods section something about p ≤ 0.05 is considered significant.
Reply: Thanks for your comments. The sentence was modified to“There was a reduction in HbA1c (-1.81%) among those with a baseline HbA1c greater than or equal to 9.”
The sentence of “p ≤ 0.05 is considered significant” was added in the methods section.
- Page 5, Line 169:In this paragraph you discuss that the efficacy of DPP-4 inhibitors and GLP-1 receptor agonists is greater if the baseline HbA1c is higher. Maybe it’s appropriate to mention here that this study only showed significantly decreased HbA1c in the patients with baseline HbA1c >9, when receiving GLP-1 RA.
Reply: Thanks for your suggestions. The revised section was added in this paragraph.
